# STATE-ONLY IMITATION WITH TRANSITION DYNAMICS MISMATCH

**Tanmay Gangwani**
Department of Computer Science
University of Illinois, Urbana-Champaign
gangwan2@illinois.edu

**Jian Peng**
Department of Computer Science
University of Illinois, Urbana-Champaign
jianpeng@illinois.edu

## ABSTRACT

Imitation Learning (IL) is a popular paradigm for training agents to achieve complicated goals by leveraging expert behavior, rather than dealing with the hardships of designing a correct reward function. With the environment modeled as a Markov Decision Process (MDP), most of the existing IL algorithms are contingent on the availability of expert demonstrations in the same MDP as the one in which a new imitator policy is to be learned. This is uncharacteristic of many real-life scenarios where discrepancies between the expert and the imitator MDPs are common, especially in the transition dynamics function. Furthermore, obtaining expert actions may be costly or infeasible, making the recent trend towards state-only IL (where expert demonstrations constitute only states or observations) ever so promising. Building on recent adversarial imitation approaches that are motivated by the idea of divergence minimization, we present a new state-only IL algorithm in this paper. It divides the overall optimization objective into two subproblems by introducing an indirection step and solves the subproblems iteratively. We show that our algorithm is particularly effective when there is a transition dynamics mismatch between the expert and imitator MDPs, while the baseline IL methods suffer from performance degradation. To analyze this, we construct several interesting MDPs by modifying the configuration parameters for the MuJoCo locomotion tasks from OpenAI Gym [1].

## 1 INTRODUCTION

In the Reinforcement Learning (RL) framework, the objective is to train policies that maximize a certain reward criterion. Deep-RL, which combines RL with the recent advances in the field of deep-learning, has produced algorithms demonstrating remarkable success in areas such as games (Mnih et al., 2015; Silver et al., 2016), continuous control (Lillicrap et al., 2015), and robotics (Levine et al., 2016), to name a few. However, the application of these algorithms beyond controlled simulation environments has been fairly modest; one of the reasons being that manual specification of a good reward function is a hard problem. Imitation Learning (IL) algorithms (Pomerleau, 1991; Ng et al., 2000; Ziebart et al., 2008; Ho & Ermon, 2016) address this issue by replacing reward functions with expert demonstrations, which are easier to collect in most scenarios.

The conventional setting used in most of the IL literature is the availability of state-action trajectories from the expert, $\tau := \{s_0, a_0, \ldots s_T, a_T\}$, collected in an environment modeled as a Markov decision process (MDP) with transition dynamics $\mathcal{T}^{\text{exp}}$. These dynamics govern the distribution over the next state, given the current state and action. The IL objective is to leverage $\tau$ to train an imitator policy in the *same* MDP as the expert. This is a severe requirement that impedes the wider applicability of IL algorithms. In many practical scenarios, the transition dynamics of the environment in which the imitator policy is learned (henceforth denoted by $\mathcal{T}^{\text{pol}}$) is different from the dynamics of the environment used to collect expert behavior, $\mathcal{T}^{\text{exp}}$. Consider self-driving cars as an example, where the goal is to learn autonomous navigation on a vehicle with slightly different gear-transmission characteristics than the vehicle used to obtain human driving demonstrations. We therefore strive

---

[1]Code for this paper is available at https://github.com/tgangwani/RL-Indirect-imitation

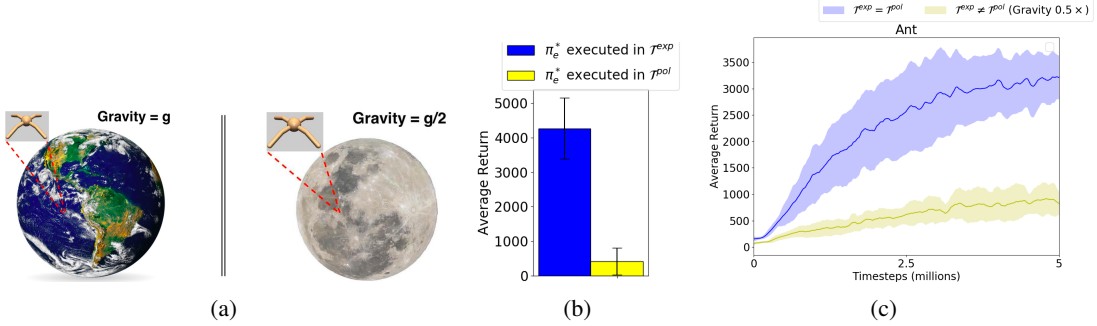

Figure 1: (a) Different amount of gravitation pull is one example of transition dynamics mismatch between the expert and the imitator MDPs. (b) An expert policy $\pi_e^*$ trained in $\mathcal{T}^{\text{exp}}$ transfer poorly to an environment with dissimilar dynamics $\mathcal{T}^{\text{pol}}$ (gravity 0.5×). (c) IL performance with GAIL degrades when $\mathcal{T}^{\text{exp}} \neq \mathcal{T}^{\text{pol}}$, compared to the conventional IL setting of imitating in the same environment as the expert.

for an IL method that could train agents under a transition dynamics mismatch, $\mathcal{T}^{\text{exp}} \neq \mathcal{T}^{\text{pol}}$. We assume that other MDP attributes are the same for the expert and imitator environments.

Beyond the dynamics equivalence, another assumption commonly used in IL literature is the availability of expert actions (along with the states). A few recent works (Torabi et al., 2018a;b; Sun et al., 2019) have proposed "state-only" IL algorithms, where expert demonstrations do not include the actions. This opens up the possibility of employing IL to situations such as kinesthetic teaching in robotics and learning from weak-supervision sources such as videos. Moreover, if $\mathcal{T}^{\text{exp}}$ and $\mathcal{T}^{\text{pol}}$ differ, then the expert actions, even if available, are not quite useful for imitation anyway, since the application of an expert action from any state leads to different next-state distributions for the expert and the imitator. Hence, our algorithm uses state-only expert demonstrations.

We build on previous IL literature inspired by GAN-based adversarial learning - GAIL (Ho & Ermon, 2016) and AIRL (Fu et al., 2017). In both these methods, the objective is to minimize the distance between the visitation distributions ($\rho$) induced by the policy and expert, under some suitable metric $d$, such as Jensen-Shannon divergence. We classify GAIL and AIRL as *direct* imitation methods as they directly reduce $d(\rho_\pi, \rho^*)$. Different from these, we propose an *indirect* imitation approach which introduces another distribution $\tilde{\rho}$ as an intermediate or indirection step. In slight detail, starting with the Max-Entropy Inverse-RL objective (Ziebart et al., 2008), we derive a lower bound which transforms the overall IL problem into two sub-parts which are solved iteratively: the first is to train a policy to imitate a distribution $\tilde{\rho}$ represented by a trajectory buffer, and the second is to move the buffer distribution closer to expert's ($\rho^*$) over the course of training. The first part, which is policy imitation by reducing $d(\rho_\pi, \tilde{\rho})$ is done with AIRL, while the second part, which is reducing $d(\tilde{\rho}, \rho^*)$, is achieved using a Wasserstein critic (Arjovsky et al., 2017). We abbreviate our approach as **I2L**, for indirect-imitation-learning.

We test the efficacy of our algorithm with continuous-control locomotion tasks from MuJoCo. Figure 1a depicts one example of the dynamics mismatch which we evaluate in our experiments. For the *Ant* agent, an expert walking policy $\pi_e^*$ is trained under the default dynamics provided in the OpenAI Gym, $\mathcal{T}^{\text{exp}} = $ Earth. The dynamics under which to learn the imitator policy are curated by modifying the gravity parameter to half its default value (i.e. $\frac{9.81}{2}$), $\mathcal{T}^{\text{pol}} = $ PlanetX. Figure 1b plots the average episodic returns of $\pi_e^*$ in the original and modified environments, and proves that direct policy transfer is infeasible. For Figure 1c, we just assume access to state-only expert demonstrations from $\pi_e^*$, and do IL with the GAIL algorithm. GAIL performs well if the imitator policy is learned in the same environment as the expert ($\mathcal{T}^{\text{exp}} = \mathcal{T}^{\text{pol}} = $ Earth), but does not succeed under mismatched transition dynamics, ($\mathcal{T}^{\text{exp}} = $ Earth, $\mathcal{T}^{\text{pol}} = $ PlanetX). In our experiments section, we consider other sources of dynamics mismatch as well, such as agent-density and joint-friction. We show that I2L trains much better policies than baseline IL algorithms in these tasks, leading to successful transfer of expert skills to an imitator in an environment dissimilar to the expert.

We start by reviewing the relevant background on Max-Entropy IRL, GAIL and AIRL, since these methods form an integral part of our overall algorithm.

## 2 BACKGROUND

An RL environment modeled as an MDP is characterized by the tuple $(\mathcal{S}, \mathcal{A}, \mathcal{R}, \mathcal{T}, \gamma)$, where $\mathcal{S}$ is the state-space, and $\mathcal{A}$ is the action-space. Given an action $a_t \in \mathcal{A}$, the next state is governed by the transition dynamics $s_{t+1} \sim \mathcal{T}(s_{t+1}|s_t, a_t)$, and reward is computed as $r_t = \mathcal{R}(r_t|s_t, a_t)$. The RL objective is to maximie the expected discounted sum of rewards, $\eta(\pi_\theta) = \mathbb{E}_{p_0, \mathcal{T}, \pi}\left[\sum_{t=0}^{\infty} \gamma^t r(s_t, a_t)\right]$, where $\gamma \in (0, 1]$ is the discount factor, and $p_0$ is the initial state distribution. We define the unnormalized $\gamma$-discounted state-visitation distribution for a policy $\pi$ by $\rho_\pi(s) = \sum_{t=0}^{\infty} \gamma^t P(s_t=s|\pi)$, where $P(s_t=s|\pi)$ is the probability of being in state $s$ at time $t$, when following policy $\pi$ and starting state $s_0 \sim p_0$. The expected policy return $\eta(\pi_\theta)$ can then be written as $\mathbb{E}_{\rho_\pi(s,a)}[r(s,a)]$, where $\rho_\pi(s,a) = \rho_\pi(s)\pi(a|s)$ is the state-action visitation distribution (also referred to as the occupancy measure). For any policy $\pi$, there is a one-to-one correspondence between $\pi$ and its occupancy measure (Puterman, 1994).

### 2.1 MAXIMUM ENTROPY IRL

Designing reward functions that adequately capture the task intentions is a laborious and error-prone procedure. An alternative is to train agents to solve a particular task by leveraging demonstrations of that task by experts. Inverse Reinforcement Learning (IRL) algorithms (Ng et al., 2000; Russell, 1998) aim to infer the reward function from expert demonstrations, and then use it for RL or planning. The IRL method, however, has an inherent ambiguity, since many expert policies could explain a set of provided demonstrations. To resolve this, Ziebart (2010) proposed the Maximum Causal Entropy (MaxEnt) IRL framework, where the objective is to learn a reward function such that the resulting policy matches the provided expert demonstrations in the expected feature counts $\mathbf{f}$, while being as random as possible:

$$\max_\pi \mathcal{H}(\pi) \quad \text{s.t.} \quad \mathbb{E}_{s,a \sim \pi}[\mathbf{f}(s,a)] = \hat{\mathbf{f}}_{\text{demo}}$$

where $\mathcal{H}(\pi) = \mathbb{E}_\pi[-\log \pi(a|s)]$ is the $\gamma$-discounted causal entropy, and $\hat{\mathbf{f}}_{\text{demo}}$ denotes the empirical feature counts of the expert. This constrained optimization problem is solved by minimizing the Lagrangian dual, resulting in the maximum entropy policy: $\pi_\theta(a|s) = \exp(Q_\theta^{\text{soft}}(s,a) - V_\theta^{\text{soft}}(s))$, where $\theta$ is the Lagrangian multiplier on the feature matching constraint, and $Q_\theta^{\text{soft}}, V_\theta^{\text{soft}}$ are the *soft* value functions such that the following equations hold (please see Theorem 6.8 in Ziebart (2010)):

$$Q_\theta^{\text{soft}}(s,a) = \underbrace{\theta^T \mathbf{f}(s,a)}_{r(s,a)} + \mathbb{E}_{p(s'|s,a)}[V_\theta^{\text{soft}}(s')] \quad , \quad V_\theta^{\text{soft}}(s) = \text{softmax}_a Q_\theta^{\text{soft}}(s,a)$$

Inspired by the energy-based formulation of the maximum entropy policy described above, $\pi_\theta(a|s) = \exp(Q_\theta^{\text{soft}}(s,a) - V_\theta^{\text{soft}}(s))$, recent methods (Finn et al., 2016; Haarnoja et al., 2017; Fu et al., 2017) have proposed to model complex, multi-modal action distributions using energy-based policies, $\pi(a|s) \propto \exp(f_\omega(s,a))$, where $f_\omega(s,a)$ is represented by a universal function approximator, such as a deep neural network. We can then interpret the IRL problem as a maximum likelihood estimation problem:

$$\max_\omega \mathbb{E}_{\tau \sim \text{demo}}[\log p_\omega(\tau)] \quad \text{with,} \quad p_\omega(\tau) = \frac{p(s_0) \prod_t p(s_{t+1}|s_t, a_t) e^{f_\omega(s_t, a_t)}}{Z(\omega)} \quad (1)$$

### 2.2 ADVERSARIAL IRL

An important implication of casting IRL as maximum likelihood estimation is that it connects IRL to adversarial training. We now briefly discuss AIRL (Fu et al., 2017) since it forms a component of our proposed algorithm. AIRL builds on GAIL (Ho & Ermon, 2016), a well-known adversarial imitation learning algorithm. GAIL frames IL as an occupancy-measure matching (or divergence minimization) problem. Let $\rho_\pi(s,a)$ and $\rho_E(s,a)$ represent the state-action visitation distributions of the policy and the expert, respectively. Minimizing the Jenson-Shanon divergence $\min_\pi D_{JS}[\rho_\pi(s,a) \| \rho_E(s,a)]$ recovers a policy with a similar trajectory distribution as the expert. GAIL iteratively trains a policy ($\pi_\theta$) and a discriminator ($D_\omega : \mathcal{S} \times \mathcal{A} \to (0,1)$) to optimize the min-max objective similar to GANs (Goodfellow et al., 2014):

$$\min_\theta \max_\omega \mathbb{E}_{(s,a) \sim \rho_E}\left[\log D_\omega(s,a)\right] + \mathbb{E}_{(s,a) \sim \pi_\theta}\left[\log(1 - D_\omega(s,a))\right] - \lambda \mathcal{H}(\pi_\theta) \quad (2)$$

GAIL attempts to learn a policy that behaves similar to the expert demonstrations, but it bypasses the process of recovering the expert reward function. Finn et al. (2016) showed that imposing a special structure on the discriminator makes the adversarial GAN training equivalent to optimizing the MLE objective (Equation 1). Furthermore, if trained to optimality, it is proved that the expert reward (up to a constant) can be recovered from the discriminator. They operate in a trajectory-centric formulation which can be inefficient for high dimensional state- and action-spaces. Fu et al. (2017) present AIRL which remedies this by proposing analogous changes to the discriminator, but operating on a single state-action pair:

$$D_\omega(s, a) = \frac{e^{f_\omega(s,a)}}{e^{f_\omega(s,a)} + \pi_\theta(a|s)} \tag{3}$$

Similar to GAIL, the discriminator is trained to maximize the objective in Equation 2; $f_\omega$ is learned, whereas the value of $\pi(a|s)$ is "filled in". The policy is optimized jointly using any RL algorithm with $\log D_\omega - \log(1 - D_\omega)$ as rewards. When trained to optimality, $\exp(f_\omega(s,a)) = \pi^*(a|s) = \exp(A^*_{\text{soft}}(s,a)/\alpha)$; hence $f_\omega$ recovers the *soft* advantage of the expert policy (up to a constant).

### 2.3 State-only Imitation

State-only IL algorithms extend the scope of applicability of IL by relieving the need for expert actions in the demonstrations. The original GAIL approach could be modified to work in the absence of actions. Specifically, Equation 2 could be altered to use a state-dependent discriminator $D_\omega(s)$, and state-visitation (instead of state-action-visitation) distributions $\rho_E(s)$ and $\rho_{\pi_\theta}(s)$. The AIRL algorithm, however, requires expert actions due to the special structure enforced on the discriminator (Equation 3), deeming it incompatible with state-only IL. This is because, even though $f_\omega$ could potentially be made a function of only the state $s$, actions are still needed for the "filled in" $\pi_\theta(a|s)$ component. Inspired by GAIL, Torabi et al. (2018b) proposed GAIfO for state-only IL. The motivation is to train the imitator to perform actions that have similar effects in the environment, rather than mimicking the expert action. Algorithmically, GAIL is modified to make the discriminator a function of state transitions $D_\omega(s, s')$, and include state-transition distributions $\rho(s, s')$.

## 3 Indirect Imitation Learning (I2L)

We now detail our I2L algorithm which alters the standard IL routine (used by GAIL, AIRL) by introducing an intermediate or *indirection* step, through a new distribution represented by a trajectory buffer. For this section, we ignore the properties of the transition dynamics for the expert and the imitator MDPs ($\mathcal{T}^{\text{exp}}, \mathcal{T}^{\text{pol}}$); they can be the same or different, I2L has no specific dependence on this. $\tau$ denotes a trajectory, which is a sequence of state-action pairs, $\{s_0, a_0, \ldots, s_T, a_T\}$. We begin with the expert's (unknown) trajectory distribution, although our final algorithm works with state-only expert demonstrations.

Let the trajectory distribution induced by the expert be $p^*(\tau)$, and its state-action visitation distribution be $\rho^*(s, a)$. Using the parameterization from Equation 1, the likelihood objective to maximize for reward learning in MaxEnt-IRL can be written as (ignoring constants *w.r.t* $\omega$):

$$\mathbb{E}_{\tau \sim p*(\tau)}[\log p_\omega(\tau)] = \mathbb{E}_{(s,a) \sim \rho^*}[f_\omega(s,a)] - \log Z(\omega) \tag{4}$$

As alluded to in Sections 2.2–2.3, *if* expert actions were available, one could optimize for $\omega$ by solving an equivalent adversarial min-max objective, as done in AIRL. To handle state-only IL, we proceed to derive a lower bound to this objective and optimize that instead. Let there be a surrogate policy $\tilde{\pi}$ with a state-action distribution $\tilde{\rho}(s, a)$. The following proposition provides a lower bound to the likelihood objective in Equation 4.

**Proposition.** *Under mild assumptions of Lipschitz continuity of the function $f_\omega$, we have that for two different state-action distributions $\rho^*$ and $\tilde{\rho}$,*

$$\mathbb{E}_{(s,a) \sim \rho^*}[f_\omega(s,a)] \geq \mathbb{E}_{(s,a) \sim \tilde{\rho}}[f_\omega(s,a)] - LW_1(\rho^*, \tilde{\rho})$$

where $L$ is the Lipschitz constant, and $W_1(\rho^*, \tilde{\rho})$ is the 1-Wasserstein (or Earth Mover's) distance between the state-action distributions.

*Proof.* Let $x := s \oplus a$ denote the concatenation of state and action. Under Lipschitz continuity assumption for $f_\omega(x)$, for any two inputs $x \sim X$ and $x' \sim X'$, we have

$$f_\omega(x') - f_\omega(x) \leq L\|(x' - x)\|_1$$

Let $\mu(X, X')$ be any joint distribution over the random variables representing the two inputs, such that the marginals are $\rho^*(X)$ and $\tilde{\rho}(X')$. Taking expectation *w.r.t* $\mu$ on both sides, we get

$$\mathbb{E}_{x' \sim \tilde{\rho}}[f_\omega(x')] - \mathbb{E}_{x \sim \rho^*}[f_\omega(x)] \leq L\mathbb{E}_\mu\|(x' - x)\|_1$$

Since the above inequality holds for any $\mu$, it also holds for $\mu^* = \arg\min_\mu \mathbb{E}_\mu\|(x' - x)\|_1$, which gives us the 1-Wasserstein distance

$$\mathbb{E}_{x' \sim \tilde{\rho}}[f_\omega(x')] - \mathbb{E}_{x \sim \rho^*}[f_\omega(x)] \leq LW_1(\rho^*, \tilde{\rho})$$

Rearranging terms,

$$\mathbb{E}_{x \sim \rho^*}[f_\omega(x)] \geq \mathbb{E}_{x' \sim \tilde{\rho}}[f_\omega(x')] - LW_1(\rho^*, \tilde{\rho})$$

$\square$

We can therefore lower bound the likelihood objective (Equation 4) as:

$$\mathbb{E}_{\tau \sim p*(\tau)}[\log p_\omega(\tau)] \geq \mathbb{E}_{\tau \sim \tilde{p}(\tau)}[\log p_\omega(\tau)] - LW_1(\rho^*, \tilde{\rho})$$

where $\tilde{p}(\tau)$ is the trajectory distribution induced by the surrogate policy $\tilde{\pi}$. Since the original optimization (Equation 1) is infeasible under the AIRL framework in the absence of expert actions, we instead maximize the lower bound, which is to solve the surrogate problem:

$$\max_{\omega, \tilde{\rho}} \mathbb{E}_{\tau \sim \tilde{p}(\tau)}[\log p_\omega(\tau)] - LW_1(\rho^*, \tilde{\rho}) \tag{5}$$

This objective can be intuitively understood as follows. Optimizing *w.r.t* $\omega$ recovers the reward (or soft advantage) function of the surrogate policy $\tilde{\pi}$, in the same spirit as MaxEnt-IRL. Optimizing *w.r.t* $\tilde{\rho}$ brings the state-action distribution of $\tilde{\pi}$ close (in 1-Wasserstein metric) to the expert's, along with a bias term that increases the log-likelihood of trajectories from $\tilde{\pi}$, under the current reward model $\omega$. We now detail the practical implementation of these optimizations.

**Surrogate policy.** We do not use a separate explicit parameterization for $\tilde{\pi}$. Instead, $\tilde{\pi}$ is implicitly represented by a buffer $\mathcal{B}$, with a fixed capacity of $k$ trajectories [2]. In this way, $\tilde{\pi}$ can be viewed as a mixture of deterministic policies, each representing a delta distribution in trajectory space. $\mathcal{B}$ is akin to experience replay (Lin, 1992), in that it is filled with trajectories generated from the agent's interaction with the environment during the learning process. The crucial difference is that inclusion to $\mathcal{B}$ is governed by a priority-based protocol (explained below). Optimization *w.r.t* $\omega$ can now be done using adversarial training (AIRL), since the surrogate policy actions are available in $\mathcal{B}$. Following Equation 3, the objective for the discriminator is:

$$\max_\omega \mathbb{E}_{(s,a) \sim \mathcal{B}}\Big[\log \frac{e^{f_\omega(s,a)}}{e^{f_\omega(s,a)} + \pi_\theta(a|s)}\Big] + \mathbb{E}_{(s,a) \sim \pi_\theta}\Big[\log \frac{\pi_\theta(a|s)}{e^{f_\omega(s,a)} + \pi_\theta(a|s)}\Big] \tag{6}$$

where $\pi_\theta$ is the learner (imitator) policy that is trained with $\log D_\omega - \log(1 - D_\omega)$ as rewards.

**Optimizing $\tilde{\rho}$.** Since $\tilde{\rho}$ is characterized by the state-action tuples in the buffer $\mathcal{B}$, updating $\tilde{\rho}$ amounts to refreshing the trajectories in $\mathcal{B}$. For the sake of simplicity, we only consider the Wasserstein distance objective and ignore the other bias term, when updating for $\tilde{\rho}$ in Equation 5. Note that $\rho^*, \tilde{\rho}$ denote the state-action visitation distributions of the expert and the surrogate, respectively. Since we have state-only demonstrations from the expert (no expert actions), we minimize the Wasserstein distance between state visitations, rather than state-action visitations. Following the approach in WGANs (Arjovsky et al., 2017), we estimate $W_1$ using the Kantorovich-Rubinstein duality, and train a critic network $g_\phi$ with Lipschitz continuity constraint,

$$W_1(\rho^*(s), \tilde{\rho}(s)) = \sup_{\|g_\phi\|_L \leq 1} \mathbb{E}_{s \sim \rho^*}[g_\phi(s)] - \mathbb{E}_{s \sim \tilde{\rho}}[g_\phi(s)] \tag{7}$$

The empirical estimate of the first expectation term is done with the states in the provided expert demonstrations; for the second term, the states in $\mathcal{B}$ are used. With the trained critic $g_\phi$, we obtain a

---

[2] $k = 5$ in all our experiments

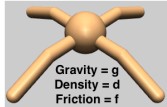 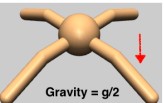 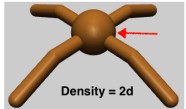 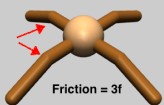

| Environment | GAIL-S | I2L | Expert (Traj. Return) |
|---|---|---|---|
| Walker2d | 3711 | **4107** | 6200 |
| Hopper | 2130 | **2751** | 3700 |
| Ant | 3217 | **3320** | 4800 |
| Half-Cheetah | **5974** | 5240 | 7500 |

Figure 2: Environments for training an imitator policy are obtained by changing the default Gym configuration settings, one at a time.

Table 1: Average episodic returns when $\mathcal{T}^{\text{exp}} = \mathcal{T}^{\text{pol}}$.

---

**Algorithm 1:** Indirect Imitation Learning (I2L)

1 Networks: Policy ($\theta$), Discriminator ($\omega$), Wasserstein critic ($\phi$)
2 $\mathcal{B} \leftarrow$ empty buffer
3 $\tau^*_{\text{states}} := \{s_0, s_1, \ldots, s_T\}$ /* State-only expert demonstration */
4 **for** *each iteration* **do**
5      Run $\pi_\theta$ in environment and collect few trajectories $\tau$
6      Update Wasserstein critic $\phi$ using $\mathcal{B}$ and $\tau^*_{\text{states}}$ /* Equation 7 */
7      Obtain trajectory score $\frac{1}{|\tau|} \sum_{s \in \tau} g_\phi(s)$ for each $\tau$ using $\phi$
8      Add $\tau$ to $\mathcal{B}$ with the priority-based protocol, using the score as priority
9      Update the AIRL discriminator $\omega$ using $\tau$ and $\mathcal{B}$ /* Equation 6 */
10      Update policy $\theta$ with PPO using $\log D_\omega - \log(1 - D_\omega)$ as rewards
11 **end**

---

*score* for each trajectory generated by the agent. The score is calculated as $\frac{1}{|\tau|} \sum_{s \in \tau} g_\phi(s)$, where $|\tau|$ is the length of the trajectory. Our buffer $\mathcal{B}$ is a priority-queue structure of fixed number of trajectories, the priority value being the score of the trajectory. This way, over the course of training, $\mathcal{B}$ is only updated with trajectories with higher scores, and by construction of the score function, these trajectories are closer to the expert's in terms of the Wasserstein metric. Further details on the update algorithm for the buffer and its alignment with the Wasserstein distance minimization objective are provided in Appendix 7.3.

**Algorithm.** The major steps of the training procedure are outlined in Algorithm 1. The policy parameters ($\theta$) are updated with the clipped-ratio version of PPO (Schulman et al., 2017). State-value function baselines and GAE (Schulman et al., 2015) are used for reducing the variance of the estimated policy-gradients. The priority buffer $\mathcal{B}$ uses the heap-queue algorithm (Appendix 7.3). The Lipschitz constant $L$ in Equation 5 is unknown and task-dependent. If $f_\omega$ is fairly smooth, $L$ is a small constant that can be treated as a hyper-parameter and absorbed into the learning rate. Please see Appendix 7.2 for details on the hyper-parameters.

## 4 RELATED WORK

There is an extensive amount of literature on IL with state-action expert demonstrations, and also on integrating IL and RL to bootstrap learning (Billard et al., 2008; Argall et al., 2009). Our work is most closely related to state-only IL and adversarial Inverse-RL methods discussed in Section 2. Here, we mention other related prior literature. BCO (Torabi et al., 2018a) is a state-only IL approach that learns an inverse dynamics model $p(a|s, s')$ by running a random exploration policy. The inverse model is then applied to infer actions from the state-only demonstrations, which in turn are used for imitation via Behavioral Cloning, making the approach vulnerable to the well-known issue of compounding errors (Ross et al., 2011). Kimura et al. (2018) learn an internal model $p(s'|s)$ on state-only demonstrations; the imitator policy is then trained with RL using rewards derived from the model. Imitation under a domain shift has been considered in Stadie et al. (2017); Liu et al. (2018). These methods incorporate raw images as observations and are designed to handle differences in context (such as viewpoints, visual appearance, object positions, surroundings) between the expert and the imitator environments. Gupta et al. (2017) propose learning invariant feature mappings to transfer skills from an expert to an imitator with a different morphology. However, the reward function for such a transfer is contingent on the assumption of time-alignment in episodic tasks. In our Algorithm 1, the adversarial training between the policy and buffer trajectories (AIRL, Line 9) bears some resemblance to the adversarial self-imitation approaches in (Guo et al., 2018; Gangwani

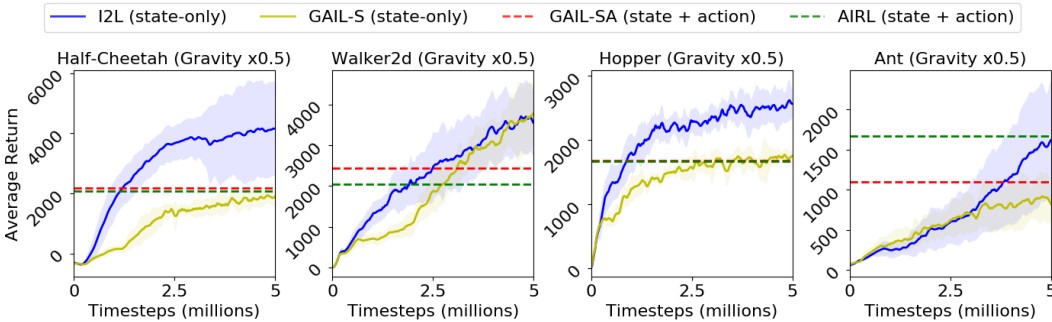

Figure 3: Training progress for I2L and GAIL-S when the imitator and expert MDPs differ in the configuration of the gravity parameter. Gravity in $\mathcal{T}^{\text{pol}}$ is $0.5\times$ the gravity in $\mathcal{T}^{\text{exp}}$.

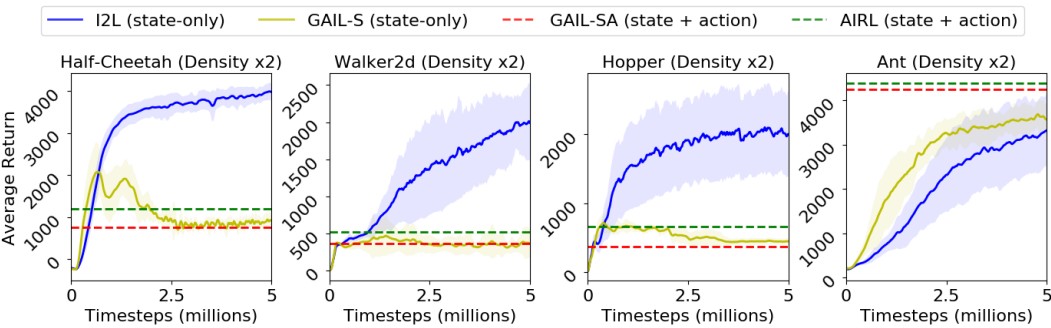

Figure 4: Training progress for I2L and GAIL-S when the imitator and expert MDPs differ in the configuration of the density parameter. Density of the bot in $\mathcal{T}^{\text{pol}}$ is $2\times$ the density in $\mathcal{T}^{\text{exp}}$.

et al., 2018). Those self-imitation methods are applicable for RL from sparse rewards, while our focus is IL from expert behavior, under transition dynamics mismatch.

## 5 EXPERIMENTS

In this section, we compare the performance of I2L to baseline methods for state-only IL from Section 2.3, namely GAIL with state-dependent discriminator, denoted by *GAIL-S*, and GAIfO (Torabi et al., 2018b). We do the evaluation by modifying the continuous-control locomotion task from MuJoCo to introduce various types of transition dynamics mismatch between the expert and the imitator MDPs ($\mathcal{T}^{\text{exp}} \neq \mathcal{T}^{\text{pol}}$). It should be noted that other aspects of the MDP ($\mathcal{S}$, $\mathcal{A}$, $\mathcal{R}$, $\gamma$) are assumed to be the same [3]. We, therefore, use dynamics and MDP interchangeably in this section. While the expert demonstrations are collected under the default configurations provided in OpenAI Gym, we construct the environments for the imitator by changing some parameters independently: a.) gravity in $\mathcal{T}^{\text{pol}}$ is $0.5\times$ the gravity in $\mathcal{T}^{\text{exp}}$, b.) density of the bot in $\mathcal{T}^{\text{pol}}$ is $2\times$ the density in $\mathcal{T}^{\text{exp}}$, and c.) the friction coefficient on all the joints of the bot in $\mathcal{T}^{\text{pol}}$ is $3\times$ the coefficient in $\mathcal{T}^{\text{exp}}$. Figure 2 has a visual. For all our experiments and tasks, we assume a *single* expert state-only demonstration of length 1000. We do not assume any access to the expert MDP beyond this.

**Performance when $\mathcal{T}^{\text{exp}} = \mathcal{T}^{\text{pol}}$.** Table 1 shows the average episodic returns for a policy trained for 5M timesteps using GAIL-S and I2L in the standard IL setting. The policy learning curves are included in Appendix 7.1. All our experiments average 8 independent runs with random seeds. Both the algorithms work fairly well in this scenario, though I2L achieves higher scores in 3 out of 4 tasks. These numbers serve as a benchmark when we evaluate performance with transition dynamics mismatch. The table also contains the expert demonstration score for each task.

---

[3]Since state-only IL does not depend on expert actions, $\mathcal{A}$ can also be made different between the MDPs without requiring any modifications to the algorithm.

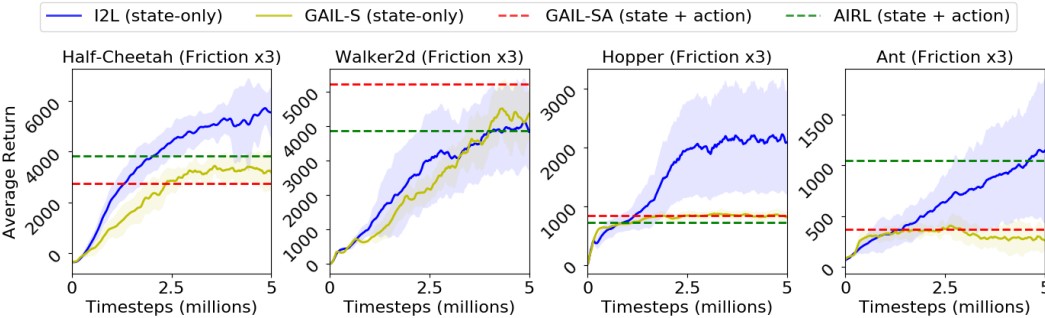

Figure 5: Training progress for I2L and GAIL-S when the imitator and expert MDPs differ in the configuration of the friction parameter. The friction coefficient on all the joints of the bot in $\mathcal{T}^{\text{pol}}$ is $3\times$ the coefficient in $\mathcal{T}^{\text{exp}}$.

**Performance when $\mathcal{T}^{\text{exp}} \neq \mathcal{T}^{\text{pol}}$.** Figures 3, 4 and 5 plot the training progress (mean and standard deviation) with GAIL-S and I2L under mismatched transition dynamics with low gravity, high density and high friction settings, respectively, as described above. We observe that I2L achieves faster

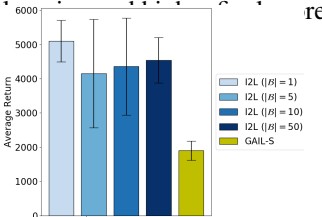

Figure 6: Ablation on capacity of buffer $\mathcal{B}$ using low-gravity *Half-Cheetah*.

| | No dynamics mismatch | | | | Low gravity | | | |
|---|---|---|---|---|---|---|---|---|
| | HalfCheetah | Walker2d | Hopper | Ant | HalfCheetah | Walker2d | Hopper | Ant |
| GAIfO | 5082 | 3122 | 2121 | 3452 | 1518 | 2995 | 1683 | 594 |
| I2L | 5240 | 4107 | 2751 | 3320 | 4155 | 3547 | 2566 | 1617 |
| | High density | | | | High friction | | | |
| | HalfCheetah | Walker2d | Hopper | Ant | HalfCheetah | Walker2d | Hopper | Ant |
| GAIfO | -234 | 378 | 440 | 3667 | 2883 | 3858 | 876 | 380 |
| I2L | 3975 | 1988 | 1999 | 3319 | 5554 | 3825 | 2084 | 1145 |

Table 2: Comparing performance of I2L with GAIfO (Torabi et al., 2018b), a state-only IL baseline.

than GAIL-S in most of the situations. GAIL-S degrades severely in some cases. For instance, for *Half-Cheetah* under high density, GAIL-S drops to 923 (compared to 5974 with no dynamics change, Table 1), while I2L attains a score of 3975 (compared to 5240 with no dynamics change). Similarly, with *Hopper* under high friction, GAIL-S score reduces to 810 (from 2130 with no dynamics change), and the I2L score is 2084 (2751 with no dynamics change). The plots also indicate the final average performance achieved using the original GAIL (marked as GAIL-SA) and AIRL algorithms. Both of these methods require extra supervision in the form of expert actions. Even so, they generally perform worse than I2L, which can be attributed to the fact that the expert actions generated in $\mathcal{T}^{\text{exp}}$ are not very useful when the dynamics shift to $\mathcal{T}^{\text{pol}}$.

**Comparison with GAIfO baseline.** GAIfO (Torabi et al., 2018b) is a recent state-only IL method which we discuss in Section 2.3. Table 2 contrasts the performance of I2L with GAIfO for imitation tasks both with and without transition dynamics mismatch. We find GAIfO to be in the same ballpark as GAIL-S. It can learn good imitation policies if the dynamics are the same between the expert and the imitator, but loses performance with mismatched dynamics. Learning curves for GAIfO are included in Appendix 7.7. Furthermore, in Appendix 7.6, we compare to BCO (Torabi et al., 2018a).

**Ablation on buffer capacity.** Algorithm 1 uses priority-queue buffer $\mathcal{B}$ of fixed number of trajectories to represent the surrogate state-action visitation $\tilde{\rho}$. All our experiments till this point fixed the buffer capacity to 5 trajectories. To gauge the sensitivity of our approach to the capacity $|\mathcal{B}|$, we ablate on it and report the results in Figure 6. The experiment is done with the low-gravity *Half-Cheetah* environment. We observe that the performance of I2L is fairly robust to $|\mathcal{B}|$. Surprisingly, even a capacity of 1 trajectory works well, and having a large buffer ($|\mathcal{B}| = 50$) also does not hurt performance much. The GAIL-S baseline on the same task is included for comparison.

**Empirical measurements of the lower-bound and Wasserstein approximations.** Section 3 introduces a lower bound on the expected value of a function $f_\omega(s, a)$ under the expert's state-action visitation. In Appendix 7.4, we analyze the quality of the lower bound by plotting the approximation-gap for the different $\tilde{\rho}$ distributions obtained during training. We observe that the gap generally re-

duces. Finally, in Appendix 7.5, we plot the empirical estimate of the Wasserstein distance between the state-visitations of the buffer distribution and the expert, and note that this value also typically decreases over the training iterations.

# 6 CONCLUSION

In this paper, we presented I2L, an *indirect* imitation-learning approach that utilizes state-only expert demonstrations collected in the expert MDP, to train an imitator policy in an MDP with a dissimilar transition dynamics function. We derive a lower bound to the Max-Ent IRL objective that transforms it into two subproblems. We then provide a practical algorithm that trains a policy to imitate the distribution characterized by a trajectory-buffer using AIRL, whilst reducing the Wasserstein distance between the state-visitations of the buffer and expert, over the course of training. Our experiments in a variety of MuJoCo-based MDPs indicate that I2L is an effective mechanism for successful skill transfer from the expert to the imitator, especially under mismatched transition dynamics.

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

# 7 APPENDIX

## 7.1 PERFORMANCE WHEN $\mathcal{T}^{\text{EXP}} = \mathcal{T}^{\text{POL}}$

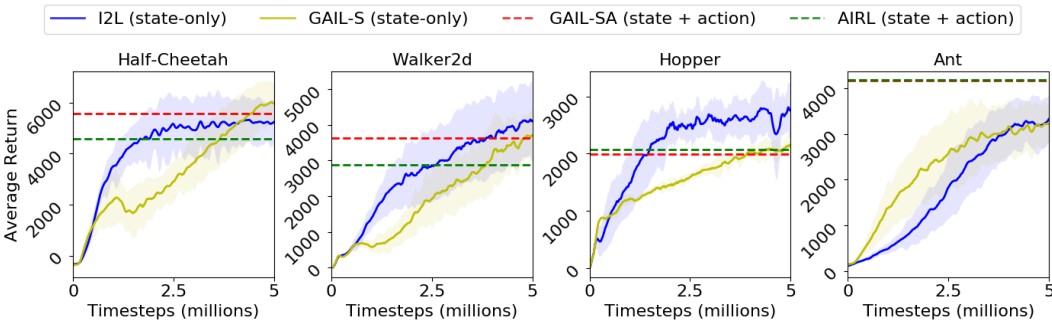

Figure 7: Training progress for I2L and GAIL-S when the imitator and expert MDPs are the same.

## 7.2 HYPER-PARAMETERS

| Hyper-parameter | Value |
|---|---|
| Wasserstein critic $\phi$ network | 3 layers, 64 hidden, tanh |
| Discriminator $\omega$ network | 3 layers, 64 hidden, tanh |
| Policy $\theta$ network | 3 layers, 64 hidden, tanh |
| Wasserstein critic $\phi$ optimizer, lr, gradient-steps | RMS-Prop, 5e-5, 20 |
| Discriminator $\omega$ optimizer, lr, gradient-steps | Adam, 3e-4, 5 |
| Policy $\theta$ algorithm, lr | PPO (clipped ratio), 1e-4 |
| Number of state-only expert demonstrations | 1 (1000 states) |
| Buffer $\mathcal{B}$ capacity | 5 trajectories |
| $\gamma, \lambda$ (GAE) | 0.99, 0.95 |

## 7.3 FURTHER DETAILS ON BUFFER $\mathcal{B}$ AND THE UPDATE MECHANISM

The buffer $\mathcal{B}$ is a priority-queue structure, with a fixed capacity of $K$ [4] trajectories. Each trajectory is a set of tuples $\tau := \{s_i, a_i\}_{i=0}^{T}$. Denote the trajectories by $\{\tau_1, \ldots, \tau_K\}$, and let $\{s \in \tau_i\}$ be the collection of states in trajectory $\tau_i$. Buffer $\mathcal{B}$ characterizes the surrogate policy $\tilde{\pi}$ defined in Section 3. The state-visitation distribution of $\tilde{\pi}$ can then be written as:

$$\tilde{\rho}(s) = \frac{1}{KT} \sum_i \sum_{s \in \tau_i} \delta(s) \tag{8}$$

where $\delta$ denotes the delta measure. Following Equation 7, our objective for optimizing $\tilde{\rho}$ is:

$$\min_{\tilde{\rho}} W_1(\rho^*(s), \tilde{\rho}(s)) \quad = \quad \min_{\tilde{\rho}} \sup_{\|g_\phi\|_L \le 1} \mathbb{E}_{s \sim \rho^*}[g_\phi(s)] - \mathbb{E}_{s \sim \tilde{\rho}}[g_\phi(s)]$$

The min-max objective is optimized using an iterative algorithm. The Wasserstein critic $g(\phi)$ update is done with standard gradient descent using state samples from expert demonstrations and the buffer $\mathcal{B}$. The update for $\tilde{\rho}$ is more challenging since $\tilde{\rho}$ is only available as an empirical measure (Equation 8). For the current iterate $\phi$, the objective for $\tilde{\rho}$ then becomes:

$$\max_{\tilde{\rho}} \mathbb{E}_{s \sim \tilde{\rho}}[g_\phi(s)] \quad = \quad \max_{\tilde{\rho}} \frac{1}{KT} \sum_i \sum_{s \in \tau_i} g_\phi(s) \tag{9}$$

Section 3 defines the quantity $\frac{1}{T} \sum_{s \in \tau_i} g_\phi(s)$ as the *score* of the trajectory $\tau_i$. Therefore, the objective in Equation 9 is to update the buffer $\mathcal{B}$ such that the average score of the $K$ trajectories in it increases.

---

[4] $K = 5$ in our experiments

**Priority-queue with priority = *score*.** Buffer $\mathcal{B}$ is implemented as a priority-queue (PQ) based on Min-Heap. Let the current PQ be $\{\tau_1, \ldots, \tau_K\}$, sorted based on *score* such that $score(\tau_i) \leq score(\tau_j)$, $\forall i < j$. Let $\{\Gamma_1, \Gamma_2, \ldots\}$ be the new trajectories rolled out in the environment using the current learner policy $\pi_\theta$ (Line 5 in Algorithm 1). For each of these, PQ is updated using standard protocol:

---

1 Update scores of $\{\tau_1, \ldots, \tau_K\}$ in PQ using the latest critic $\phi$
2 **for** *each* $\Gamma_i$ **do**
3    Calculate $score(\Gamma_i)$ using the latest critic $\phi$
4    **if** $score(\Gamma_i) > score(\tau_1)$ **then**
5       $\tau_1 \leftarrow \Gamma_i$      // replace the lowest scoring buffer trajectory with the new trajectory
6       *heapify*      // PQ-library call to maintain the heap-invariant: $score(\tau_i) \leq score(\tau_j)$, $\forall i < j$
7    **end**
8 **end**

---

It follows from the PQ-protocol that the average score of the $K$ trajectories in the buffer $\mathcal{B}$ increases (or remains same), after the update, compared to the average score before. This aligns the update with the objective in Equation 9.

## 7.4 EMPIRICAL CONVERGENCE OF LOWER BOUND

In our main section, we derive the following lower bound which connects the expected value of a function $f_\omega$ under the expert's state-action visitation to the expected value under another surrogate distribution, and the 1-Wasserstein distance between the distributions:

$$\mathbb{E}_{(s,a)\sim\rho^*}[f_\omega(s,a)] \geq \mathbb{E}_{(s,a)\sim\tilde{\rho}}[f_\omega(s,a)] - LW_1(\rho^*, \tilde{\rho})$$

In this section, we provide empirical measurements on the *gap* between the original objective (LHS) and the lower bound (RHS). This gap depends on the specific choice of the surrogate distribution $\tilde{\rho}(s,a)$. In our algorithm, $\tilde{\rho}$ is characterized by trajectories in the priority-queue buffer $\mathcal{B}$, and is updated during the course of training based on the protocol detailed in Appendix 7.3. Figure 8 plots the estimated value of the lower bound for these different $\tilde{\rho}$, and shows that the gap generally reduces over time. To get estimates of LHS and RHS, we need the following:

- $\tilde{\rho}(s,a)$ : We take snapshots of the buffer $\mathcal{B}$ at periodic intervals of the training to obtain the different $\tilde{\rho}$ distributions.

- $\rho^*(s,a)$ : This is the expert's state-action distribution. We train separate oracle experts in the imitator's (learner's) environment, and use state-action tuples from this expert policy. Note that these oracle experts are **NOT** used in I2L (Algorithm 1), and are only for the purpose of measurement.

- $W_1(\rho^*(s,a), \tilde{\rho}(s,a))$ : A separate Wasserstein critic is trained using $(s,a)$ tuples from the oracle experts described above and the trajectories in buffer $\mathcal{B}$. This critic is **NOT** used in I2L since we don't have access to oracle experts, and is only for the purpose of measurement.

- $f_\omega$ : We select the AIRL discriminator parameters $\omega$ from a particular training iteration. The same parameters are then used to calculate the LHS, and the RHS for different $\tilde{\rho}$ distributions.

- $L$ : The Lipschitz constant is unknown and hard to estimate for a complex, non-linear $f_\omega$. We plot the lower bound for a few values: $\{0.5, 1.0, 1.5\}$.

Figure 8 shows the gap between the original objective and the lower bound for all our experimental settings, $\mathcal{T}^{\text{exp}} = \mathcal{T}^{\text{pol}}$ (top row), and $\mathcal{T}^{\text{exp}} \neq \mathcal{T}^{\text{pol}}$ (next 3 rows). We observe that the gap generally reduces as $\tilde{\rho}$ is updated over the iterations of I2L (Algorithm 1). A better lower bound in turn leads to improved gradients for updating the AIRL discriminator $f_\omega$, ultimately resulting in more effective policy gradients for the imitator.

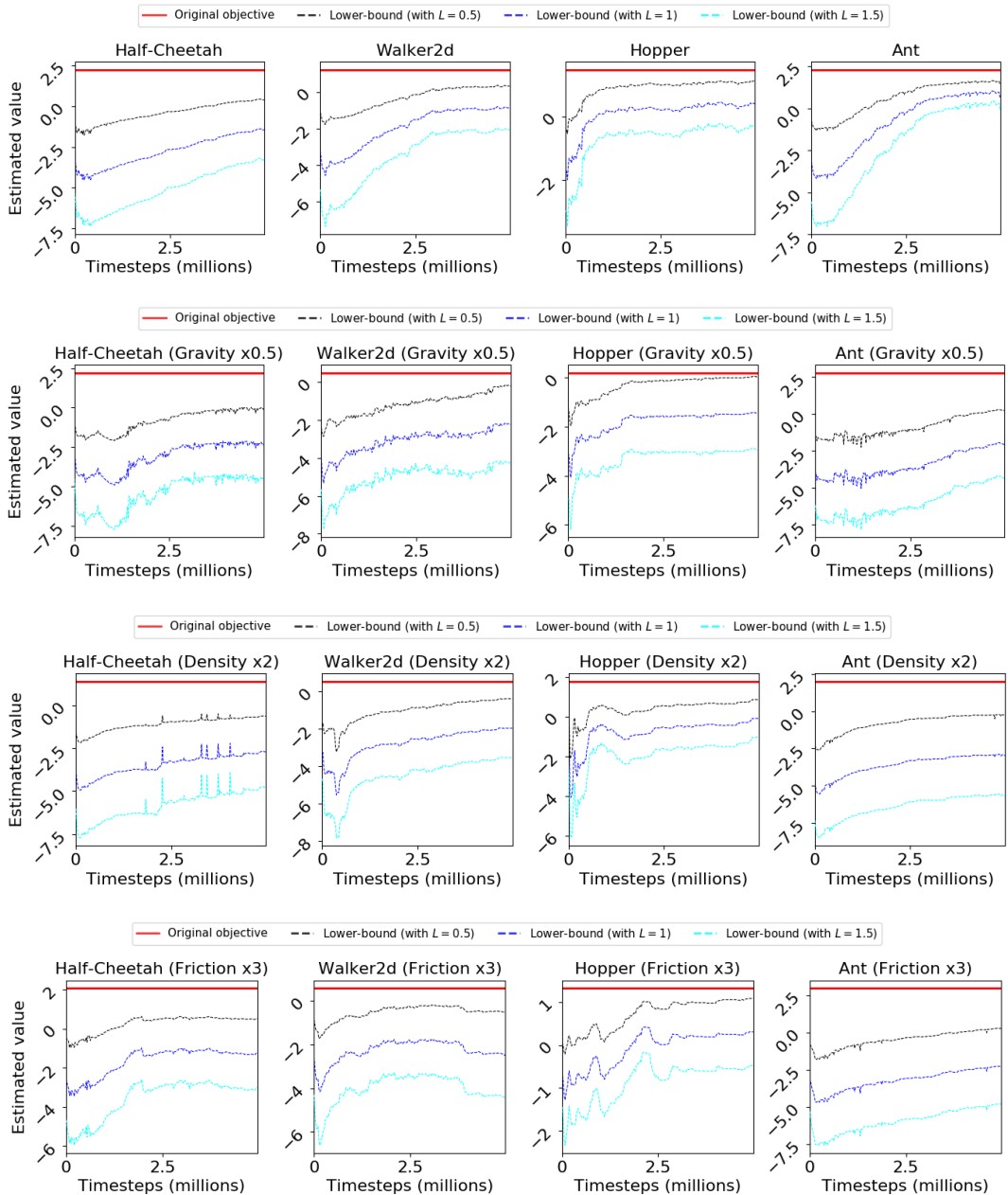

Figure 8: Gap between the original objective and the lower bound for all our experimental settings, $\mathcal{T}^{\mathrm{exp}} = \mathcal{T}^{\mathrm{pol}}$ (top row), and $\mathcal{T}^{\mathrm{exp}} \neq \mathcal{T}^{\mathrm{pol}}$ (next 3 rows).

## 7.5 EMPIRICAL WASSERSTEIN DISTANCES

In each iteration of I2L, we update the Wasserstein critic $\phi$ using the states from the state-only expert demonstration, and states in the buffer $\mathcal{B}$ (Line 6, Algorithm 1). The objective is to obtain the 1-Wasserstein distance:

$$W_1\big(\rho^*(s), \tilde{\rho}(s)\big) = \sup_{\|g_\phi\|_L \leq 1} \mathbb{E}_{s \sim \rho^*}[g_\phi(s)] - \mathbb{E}_{s \sim \mathcal{B}}[g_\phi(s)]$$

In Figure 9, we plot the empirical estimate $\hat{W}_1$ of this distance over the course of training. To get the estimate at any time, the current critic parameters $\phi$ and buffer trajectories are used to calculate $\hat{E}_{s \sim \rho^*}[g_\phi(s)] - \hat{E}_{s \sim \mathcal{B}}[g_\phi(s)]$. We show the values for all our experimental settings, $\mathcal{T}^{\mathrm{exp}} = \mathcal{T}^{\mathrm{pol}}$

(top row), and $\mathcal{T}^{\mathrm{exp}} \neq \mathcal{T}^{\mathrm{pol}}$ (next 3 rows). It can be seen that $\hat{W}_1(\rho^*(s), \tilde{\rho}(s))$ generally decreases over time in all situations. This is because our objective for optimizing $\tilde{\rho}$ (or updating the buffer $\mathcal{B}$) is to minimize this Wasserstein estimate. Please see Appendix 7.3 for more details. The fact that the buffer state-distribution gets closer to the expert's, together with the availability of actions in the buffer which induce those states in the imitator MDP ($\mathcal{T}^{\mathrm{pol}}$), enables us to successfully use AIRL for imitation under mismatched transition dynamics.

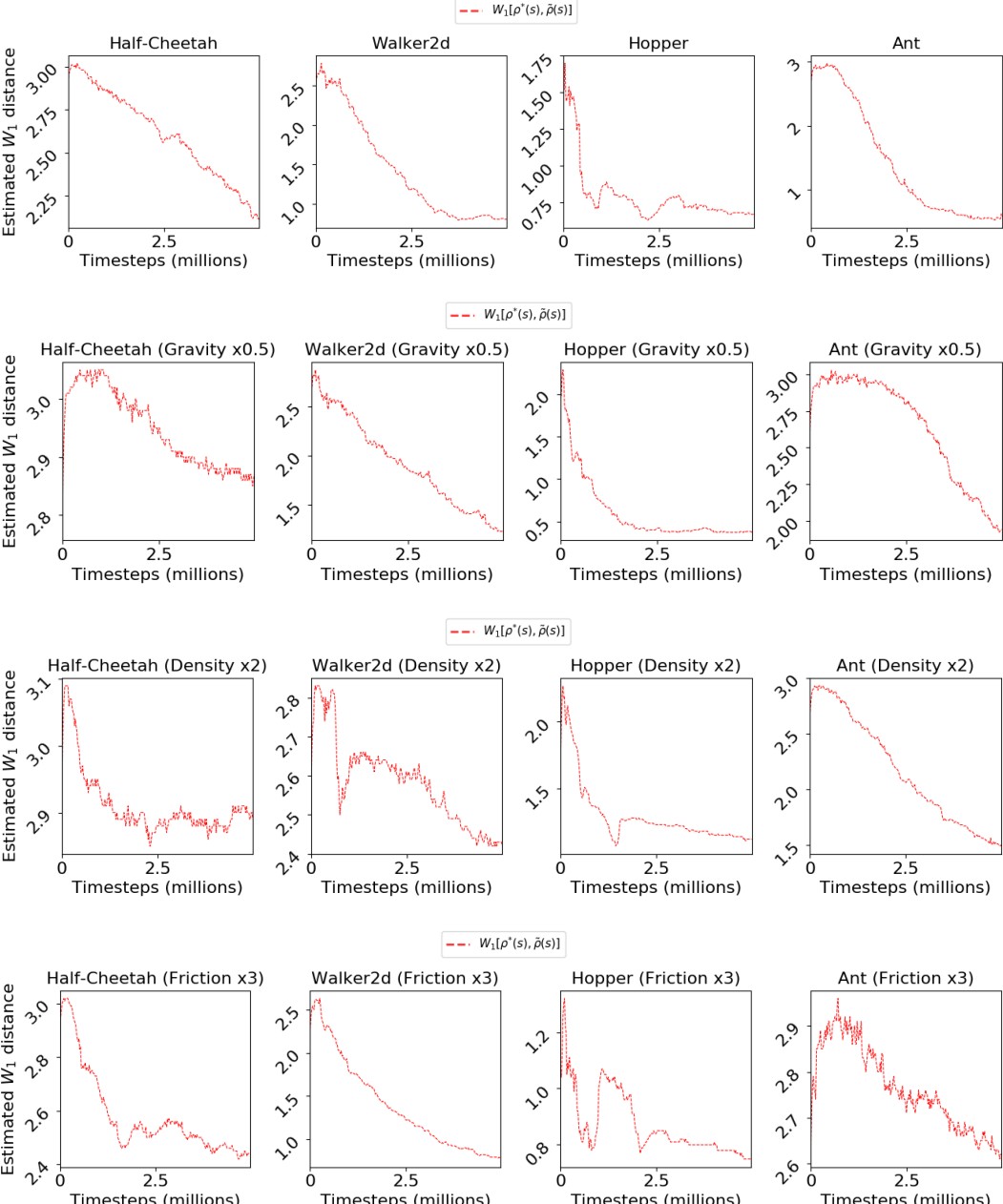

Figure 9: Estimate of the Wasserstein distance between $\rho^*(s)$ and $\tilde{\rho}(s)$, for all our experimental settings, $\mathcal{T}^{\mathrm{exp}} = \mathcal{T}^{\mathrm{pol}}$ (top row), and $\mathcal{T}^{\mathrm{exp}} \neq \mathcal{T}^{\mathrm{pol}}$ (next 3 rows).

## 7.6 COMPARISON WITH BCO

Figure 10 compares I2L with BCO (Torabi et al., 2018a) when the expert and imitator dynamics are same (top row) and under mismatched transition dynamics with low gravity, high density and high

friction settings (next 3 rows). BCO proceeds by first learning an inverse dynamics model in the imitator's environment, $p(a|s, s')$, to predict actions from state-transitions. This model is learned via supervised learning on trajectories generated by an exploratory policy. The inverse model is then used to infer actions from the state-transitions in state-only expert demonstrations. The imitator policy is trained with Behavioral Cloning (BC) using these inferred actions. We implement the BCO($\alpha$) version from the paper since it is shown to be better than vanilla BCO. We observe the barring two situations (Ant with no dynamics mismatch, and Ant with $2\times$ density), BCO($\alpha$) is unsuccessful in learning high-return policies. This is potentially due to the difficulties in learning a robust inverse dynamics model, and the compounding error problem inherent to BC. Similar performance for BCO($\alpha$) is also reported by Torabi et al. (2018b).

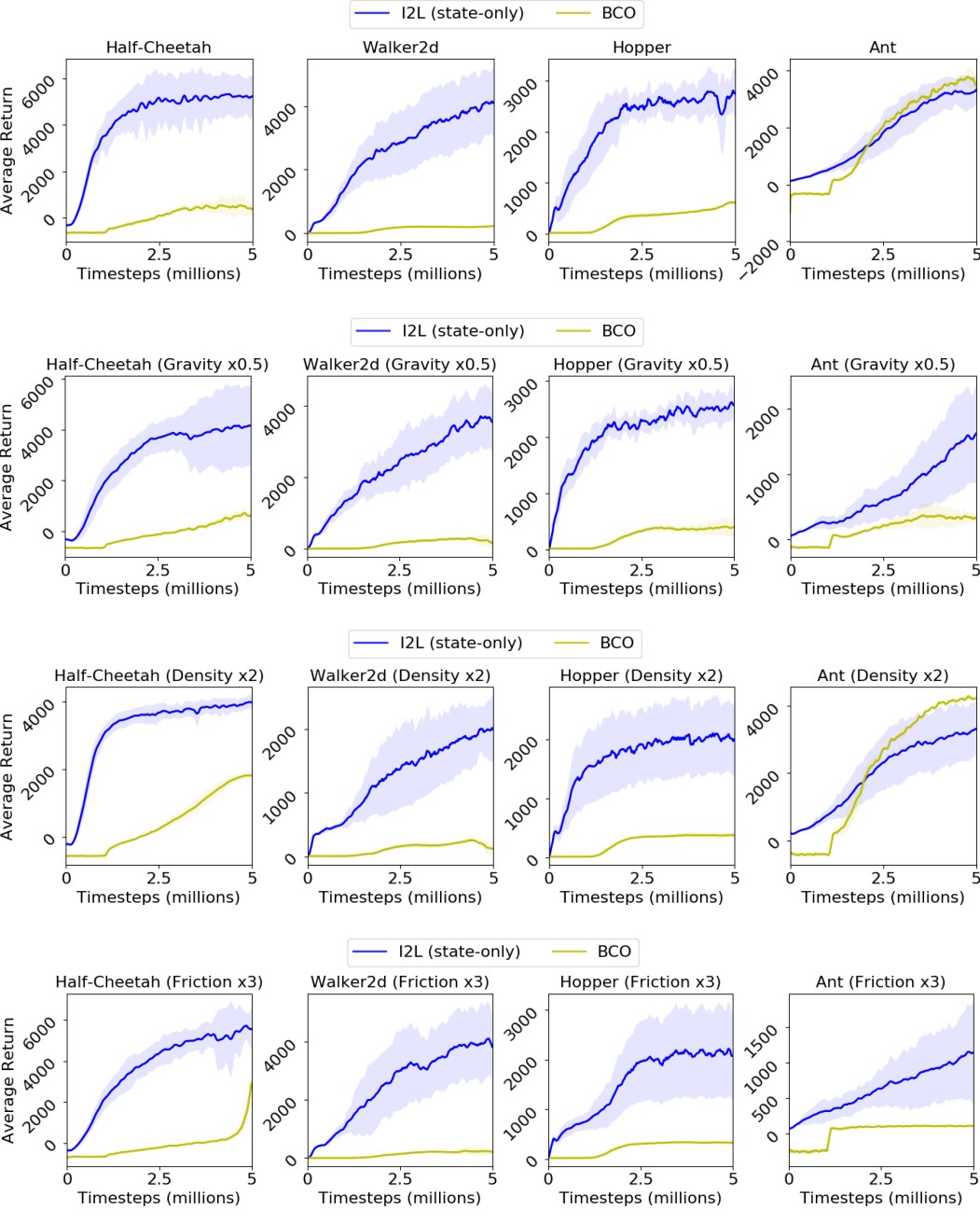

Figure 10: Comparison between I2L and BCO.

## 7.7 COMPARISON WITH GAIFO

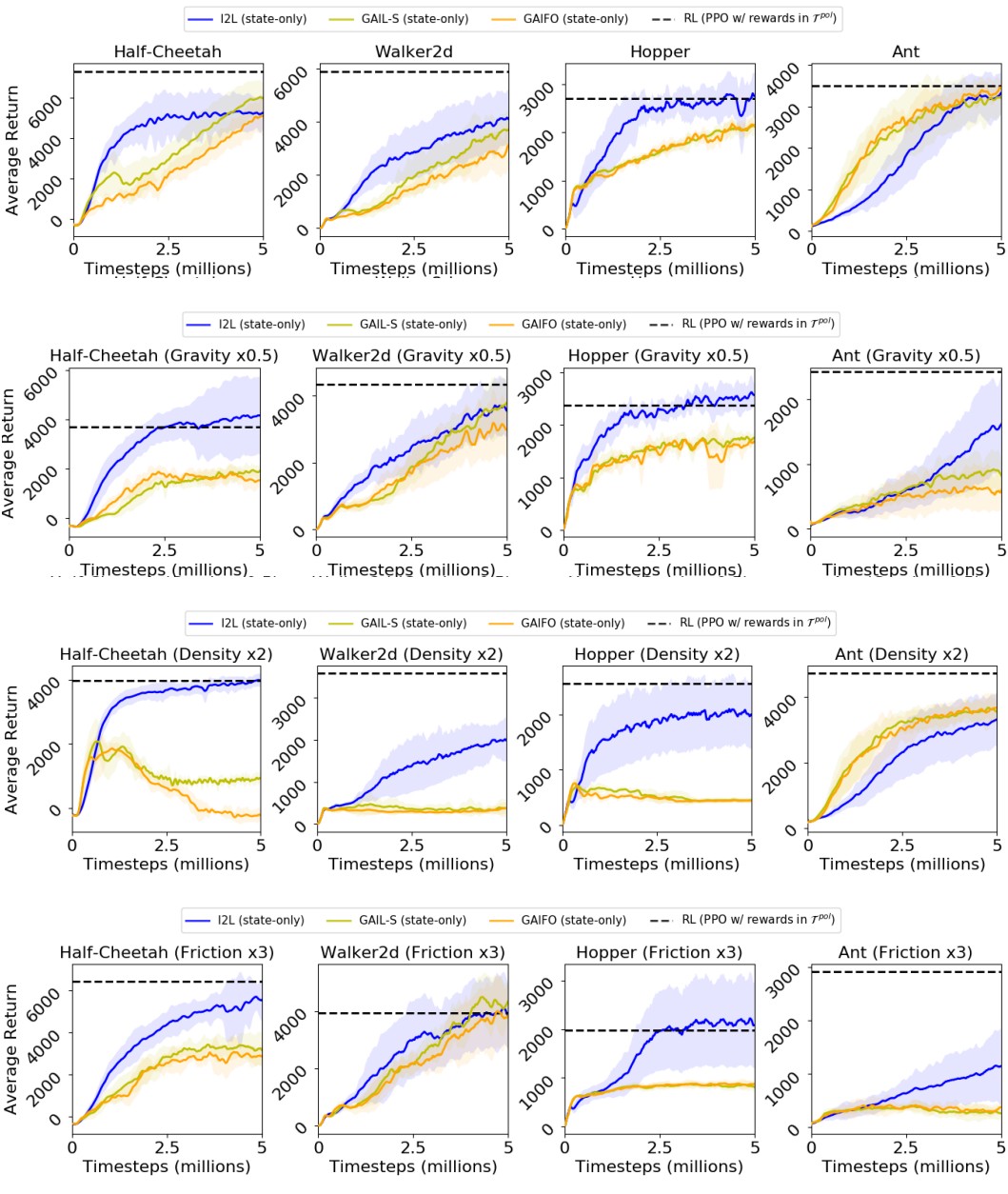

Figure 11: Comparison between I2L and two baselines derived from GAIL. The final performance of an agent trained with PPO using real rewards in $\mathcal{T}^{\text{pol}}$ is also shown.

## 7.8 COMPARISON WITH GAIL-SA AND AIRL

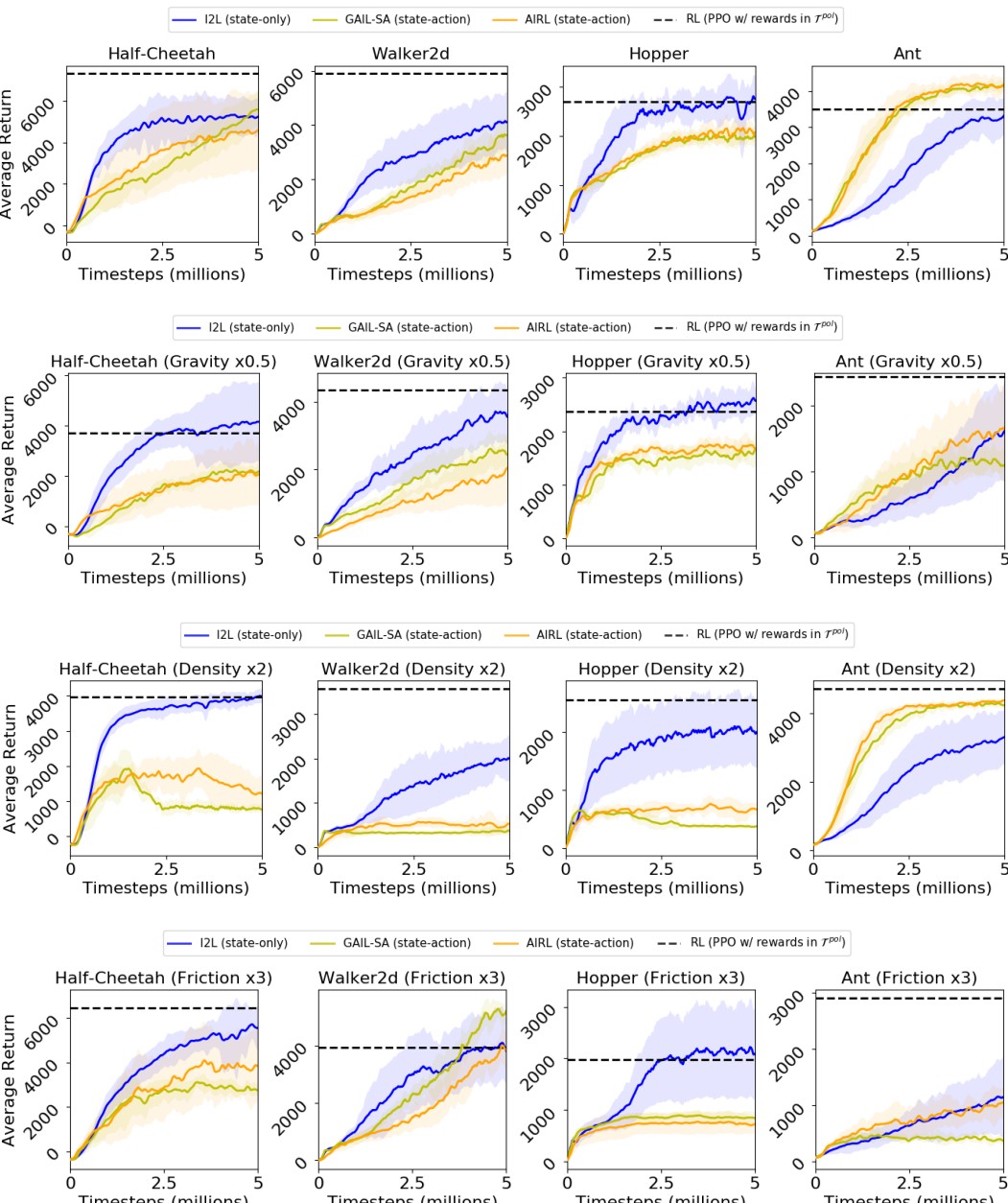

Figure 12: Comparison between I2L and baselines using expert actions: GAIL-SA and AIRL

