# OpenReview forum: "State-only Imitation with Transition Dynamics Mismatch"
_ICLR.cc/2020/Conference — Accept (Poster)_

### Official Review · AnonReviewer1 · 2019-10-22
**Official Blind Review #1**

**Rating:** 6

**Review:**

Summary:
The manuscript considers the problem of imitation learning when the system dynamics of the agent are different from the dynamics of the expert. The paper proposes Indirect Imitation Learning (I2L), which aims to perform imitation learning with respect to a trajectory buffer that contains some of the previous trajectories of the agent. The trajectory buffer has limited capacity and adds trajectories based on a priority-queue that prefers trajectories that have a similar state distribution to the expert. Similarity is hereby measured by the score of a WGAN-critic trained to approximate the W1-Wasserstein distance between the previous buffer and the expert distribution. By performing imitation learning with respect to a trajectory buffer, state-action trajectories of the agent's MDP are available, which enables I2L to apply AIRL (Fu et al. 2017). By using those trajectories for the transition buffer that have state-marginals close to the expert's trajectory, I2L produces similar behavior compared to the expert. I2L is compared to state-only GAIL, state-action-GAIL and AIRL on four MuJoCo tasks with modified dynamics compared to the expert policy. The experiments show that I2L may learn significantly better policies if the dynamics of agent and the expert do not match.

Decision:
I think that the submission is below borderline in its current state. The main reason for rejecting would be the insufficient justification of some algorithmic choices, improper presentation of the experiments and the description of MaxEnt-IRL, which seems quite wrong in my opinion. However, I think that the work is interesting and sufficiently novel and could be accepted if the mentioned issues were adequately addressed.

Supporting Arguments:
- Novelty / Significance: Imitation learning from state-only observations under dynamic mismatch is an important problem and a promising approach for training robots by non-experts. The idea of performing imitation learning with respect to carefully updated trajectory buffer seems simple and effective. Although self-imitation has been applied in reinforcement learning (references in submission), I am not aware of a similar approach in imitation learning.

- Soundness / Correctness:
1. The description of MaxEnt-IRL seems quite wrong. The paper claims in Section 2.1. that MaxEnt-IRL maximizes the likelihood of the policy by optimizing the advantage function and thus only learns shaped rewards. However, the referenced paper (Ziebart et al. 2008) optimizes the weights of a linear reward function which in general does not correspond to the advantage of the learned policy. Also, the policy is not (only) proportional to the exponentiated advantage but equal to it and the normalizer of the trajectory distribution (Eq. 1) would be therefore 1.

2. I think that the selection of trajectories for the buffer is not sufficiently well motivated. I2L is derived based on a lower bound but it seems that it does not even ensure improvement on that lower bound. It is not fully clear to me how the trajectory buffer is updated (see "Clarity") but it does not seem to ensure that the Wasserstein distance decreases compared to the last iteration. The trajectories are chosen greedily, i.e., without considering the remaining trajectories in the buffer which can be especially problematic for multimodal demonstrations.

- Clarity / Presentation:
The paper is well-written in general and only has few typos. It is not clear to me how exactly the trajectory buffer is updated. The submission merely states that the buffer is a priority-queue structure of fixed lengths, where the trajectory priorities are given by the average state score based on the Wasserstein critic. This description leaves several open questions (see "Questions") and further does not seem well motivated.


- Evaluation:
The presentation of the experimental results seems odd. For the imitation learning experiments (no dynamic mismatch) the (apparently) same runs of I2L are compared to GAIL-S (Table 1) and GAIfO (Table 2) in separate tables. For the experiments under dynamic mismatch, the comparison with GAIL-S are shown with learning curves (Figure 3-5) and shaded error bars, whereas the results of GAIfO are only shown in in terms of final mean performance in Table 2. This presentation may make the impression or hiding learning curves or confidence intervals. Both tables could be removed by adding a single curve to each of the plots in Figure 3-5 and 7. It is also not clear to me, why only the mean of the final performance for AIRL and SA-GAIL is shown in Figure 3-5. Instead, the performance of the expert--which is currently not shown--would be better suited as a baseline level. The paper also does not seem to mention the number of evaluations and the meaning of the error region in the figures. Furthermore, some hyper-parameters, e.g., the architectures for the AIRL reward/value-function networks are not presented.

Questions:
- Please precisely state when the trajectories are removed from the trajectory buffer and the heap of the priority queue. May the same trajectories be used during several iterations?
- Please elaborate: under which assumptions does the update of the trajectory buffer ensure that the W1-distance of the individual trajectories decreases with respect to the expert's trajectories?


Minor comments:
Typo: "upto a constant"

Post-Rebuttal Update:
I increased my rating to weak accept because the authors addressed my main concerns by a) giving additional information on the update of the buffer, b) improving the presentation of the experiments, c) fixing the description of MaxEnt-IRL and d) by showing empirically that both the lower bound increases / Wasserstein-distance decreases.

**Experience Assessment:**

I have published one or two papers in this area.

**Review Assessment: Checking Correctness Of Derivations And Theory:**

I assessed the sensibility of the derivations and theory.

**Review Assessment: Checking Correctness Of Experiments:**

I assessed the sensibility of the experiments.

**Review Assessment: Thoroughness In Paper Reading:**

I read the paper thoroughly.

---

> ### Author Response · Authors · 2019-11-15
> **[Part 1 of 2] Response to AnonReviewer1. Thank you for your comments!**
>
>
> 1- Regarding “Lack of clarity on the update mechanism of the buffer”
>
> Thank you for expressing this concern. We acknowledge that we may have condensed a bit too much when describing the buffer update in Section 3. We have now added Appendix 7.3 with further details on the update mechanism of the buffer. It mathematically defines the buffer distribution in terms of delta measures, and from there motivates using the trajectory ‘score’ (as defined in our main section) as a way to reduce the Wasserstein distance between expert-states and buffer-states. For completeness, we also outline the min-heap based priority queue algorithm. We would be happy to include any further details the reviewer may suggest.
>
>
> 2- Regarding “I2L is derived based on a lower bound but it seems that it does not even ensure improvement on that lower bound”
>
> We have added Appendix 7.4 for this. The figures therein plot that gap between the original objective and lower bound, for all the environmental settings considered in the paper, and show that the gap generally reduces as the buffer distribution is updated over the iterations of the I2L algorithm. Please see our description and plots in Appendix 7.4, and also our response to AnonReviewer3.
>
>
> 3- Regarding “It does not seem to ensure that the Wasserstein distance decreases compared to the last iteration”
>
> We have added Appendix 7.5 with plots which show the estimated Wasserstein distance between expert and buffer (state) distributions, over the course of training. We observe that this value generally decreases over time, for all the environmental settings considered in the paper. Please see our description and plots in Appendix 7.5.
>
>
> 4- Regarding “Please precisely state when the trajectories are removed from the trajectory buffer and the heap of the priority queue. May the same trajectories be used during several iterations?”
>
> We hope that the details in Appendix 7.3 provide a clearer picture of when and how the buffer trajectories are updated. In summary, in each iteration of the I2L algorithm (Algorithm 1 in the paper), line 8 is where the update to the buffer happens. The exact rules for the update are in Appendix 7.3 (the algorithm box therein). We also motivate the update rule there.
>
>
> 5- Regarding "Please elaborate: under which assumptions does the update of the trajectory buffer ensure that the W1-distance of the individual trajectories decreases with respect to the expert's trajectories?"
>
> The update mechanism for the buffer is designed such that empirical 1-Wasserstein distance between expert and buffer (state) distributions reduces. Specifically, using a priority-queue buffer, and adding trajectories using ‘score’ as the priority ensures the reduction. This is mathematically shown in Appendix 7.3 and empirically in Appendix 7.5
>
>
> continued below...

---

> > ### Author Response · Authors · 2019-11-15
> > **[Part 2 of 2] Response to AnonReviewer1. Thank you for your comments!**
> >
> >
> > continued...
> >
> > 6- Regarding “Description of MaxEnt-IRL using advantage does not seem correct”
> >
> > We acknowledge that this might cause some confusion in the reader’s mind, and therefore, we have added a paragraph in Section 2.1 to explain our parameterization (Please see the text in blue in Section 2.1). Our approach for directly parameterizing advantage is motivated by the fact that we use AIRL as our MaxEnt-IRL method; AIRL recovers the advantage function of the expert. But we explain that other parameterizations are possible. Further justification for using the advantage is also provided.
> >
> > Concerning the normalizer when using parametric advantage, it would be 1 only when $a_w$ “converges” to the actual advantage, which is the optimal point of the AIRL min-max training. For a general parameterization (without external constraints), we need to have the normalizer, which is a function of $w$.
> >
> >
> > 7- Regarding “Presentation of experiments”
> >
> > Learning curves for GAIfO and unnecessary tables - We had added the GAIfO learning curves in Appendix 7.7. We appreciate this suggestion on improving the presentation of our results. We are going to remove the unnecessary tables from the final manuscript and replace them with plots from Appendix 7.7
> >
> > Why are AIRL and SA-GAIL shown - They were added to show that while these methods work well when the dynamics match, the performance degrades under dynamics mismatch. A less important point was to convince the reader that we have implemented the baseline methods to the best of our ability, as shown by the good performance in the same dynamics scenario. We can move these to the Appendix.
> >
> > Show performance of experts in plots - This is now included in the plots in Appendix 7.7. We use PPO with the real environmental rewards for each of our MDPs, and plot the final performance of these “oracle” experts. We stress that we have state-only demonstrations *only* from the expert in the unmodified MuJoCo MDP. The oracle experts for modified environments (changed gravity, density, friction) are only to get the final performance for the purpose of plotting.
> >
> > “Number of evaluations” - This was mentioned in the second paragraph of Section 5. All our experiments average 8 independent runs with random seeds.
> >
> > “Meaning of the error region” - Thanks for pointing this out. This has been rectified. We plot the mean and standard deviation.
> >
> > “AIRL reward/value-function network architecture” - These were already mentioned in Appendix 7.2. We would be happy to provide more details if required.

---

> > > ### Comment · AnonReviewer1 · 2019-11-15
> > > **Thank you for the revision**
> > >
> > > Thank you for the revision. I think some parts have been clearly improved.
> > > It is now clear how exactly the buffer is updated, and the empirical results showing the improvement of the lower of the lower bound / Wasserstein-distance are assuring.
> > >
> > > However, some of my comments / question are not yet adequately addressed:
> > >
> > > 1) Regarding the depiction of MaxEnt-IRL:
> > > Please note that MaxEnt-IRL is a specific algorithm proposed my Ziebart et al. (2008) with the following properties:
> > > - Based on the principle of maximum entropy it aims to find the MaxEnt distribution that matches given, observed features in expectation.
> > > - It learns a reward function that is linear in these features. This form of reward function is _not_ a design choice! The weights correspond to the Lagrangian multiplier w.r.t. the feature matching constraint. MaxEnt-IRL is _not_ ill-posed, the Lagrangian dual is convex in the weights.
> > >
> > > Yes, there a several methods that, often inspired by the MaxEnt-IRL dual, aim to maximize the likelihood of a Boltzmann-policy. These methods may not be well-posed and may further be hard to justify by the principle of MaxEnt. I understand that you use MaxEnt-IRL as umbrella term for these method. However, you need to understand that using the name and reference of MaxEnt-IRL and claiming that it parametrizes the advantage function and that it is ill-posed by having some "second ambiguity" produces a completely wrong depiction of that algorithm.
> > >
> > > 2) I did not ask why AIRL and SA-GAIL are shown. I asked why _only the mean_ is shown.
> > >
> > > 3) I still can not see the parametrization of the value-function inside the AIRL-discriminator.

---

> > > > ### Author Response · Authors · 2019-11-15
> > > > **Thank you for further feedback.**
> > > >
> > > >
> > > > 1- "Depiction of MaxEnt-IRL"
> > > >
> > > > Our corrections to the background on MaxEnt-IRL gave the wrong impression that MaxEnt makes design choice assumptions on the reward function. Our description also somewhat blurred the lines between the original MaxEnt-IRL formulation and recent works using maximum likelihood with energy-based policies. We agree with the reviewer that it’s very important to be crisp about established principles like MaxEnt-IRL, and clearly outline the transition to algorithms that are *inspired* by MaxEnt-IRL, such as AIRL.
> > > >
> > > > We have strived to do this by modifying Section 2.1. Please see the changed text in blue. We describe the MaxEnt-IRL formulation in terms of the feature matching constraint and mention the solution to the optimization problem as per [1]. We then allude to recent methods [2-5] which are motivated by maximum entropy policies and employ an energy-based distribution, with energy parameterized by a neural network.
> > > >
> > > > We are very grateful to the reviewer for their help on this subtle issue.
> > > >
> > > > We would also like to present some clarification on reference to the “second ambiguity”. We did not mean to say that the “second ambiguity” is specific to MaxEnt-IRL. The IRL framework, in general, has the following 2 ambiguities, also brought up in recent works such as [4,5]:
> > > >
> > > >   a. Many expert policies could explain a set of provided demonstrations
> > > >   b. Many reward functions lead to the same optimal policy [Reward shaping ambiguity]
> > > >
> > > > [1] Brian D Ziebart. Modeling purposeful adaptive behavior with the principle of maximum causal entropy
> > > > [2] Finn et al. A connection between generative adversarial networks, inverse reinforcement learning, and energy-based models
> > > > [3] Haarnoja et al. Reinforcement learning with deep energy-based policies
> > > > [4] Fu et al. Learning robust rewards with adversarial inverse reinforcement learning
> > > > [5] Yu et al. Multi-Agent Adversarial Inverse Reinforcement Learning
> > > >
> > > >
> > > > 2- “I still can not see the parametrization of the value-function inside the AIRL-discriminator”
> > > >
> > > > We do not have separate reward and value (V) networks inside the AIRL discriminator. A single network--with 3 layers of 64 hidden units and tanh nonlinearity--is used to parameterize the advantage (or shaped rewards). If the reviewer is alluding to the original AIRL paper with separate reward and value networks to learn *disentangled rewards*, then we would like to note that the separate parameterization there serves a very specific purpose -- which is to learn rewards that are not shaped by environment dynamics, and can therefore be used for other related tasks/environments. For our paper, the goal is to learn a performant policy in the imitator MDP using demonstrations from a different expert MDP. Our rewards for the imitator could be shaped by the imitator MDP; we are not concerned about that aspect. Of course, it is straightforward to include a separate reward and value parameterization in our method as well, but we found the single network to work sufficiently well.
> > > >
> > > >
> > > > 3- AIRL and SA-GAIL plots
> > > > These are now provided in Appendix 7.8. We would combine all plots in the final revision.

---

### Official Review · AnonReviewer3 · 2019-10-22
**Official Blind Review #1**

**Rating:** 6

**Review:**

The submission considers the imitation learning with environment change. In the new environment (where we we aim to conduct imitation learning), the expert demonstrations are unavailable, making the objective function (5) unavailable. To deal with this, the authors derive a lower bound to replace (5).

In the experiment section, the performance of the proposed method is clearly demonstrated. Environment change is a challenging case for existing imitation learning methods, while the proposed one works.

While empirically, the performance of the proposed method is justified, I am curious how tight the bound used to replace (or approximate) (5) is. The submission offers few discussions on the error may induced by replacing (5) with the lower bound. Will such an error decrease or converge to 0, as we iterate according to the algorithm?

**Experience Assessment:**

I have read many papers in this area.

**Review Assessment: Checking Correctness Of Derivations And Theory:**

I assessed the sensibility of the derivations and theory.

**Review Assessment: Checking Correctness Of Experiments:**

I assessed the sensibility of the experiments.

**Review Assessment: Thoroughness In Paper Reading:**

I read the paper at least twice and used my best judgement in assessing the paper.

---

> ### Author Response · Authors · 2019-11-14
> **Response to AnonReviewer3. Thank you for your comments!**
>
>
> 1- Regarding question on the error of the lower bound.
>
> We thank the reviewer for this question, for it motivated us to do some empirical analysis on the convergence of the lower bound to the original likelihood objective (Proposition in the paper). We have added Appendix 7.4 in the revision with our observations. Therein, we detail how we estimate the values of the original objective and the lower bound for different buffer distributions. The figures in Appendix 7.4 plot that gap between the original objective and lower bound, for all the environmental settings considered in the paper, and show that the gap generally reduces as the buffer distribution is updated over the iterations of the I2L algorithm. A better lower bound in turn leads to improved gradients for updating the AIRL discriminator, ultimately resulting in more effective policy gradients for the imitator. We also believe that the lower bound we obtained in this proposition, although quite simple to prove, is non-trivial. For example, if the function approximation ($a_w$) is linear, the lower bound becomes tight and cannot be further improved.

---

### Official Review · AnonReviewer2 · 2019-10-23
**Official Blind Review #2**

**Rating:** 6

**Review:**

The paper proposes an imitation method, I2L, that learns from state-only demonstrations generated in an expert MDP that may have different transition dynamics than the agent MDP. I2L modifies the existing adversarial inverse RL algorithm: instead of training the disciminator to distinguish demonstrations vs. samples, I2L trains the discriminator to distinguish samples that are close (in terms of the Wasserstein metric) to the demonstrations vs. other samples. This approach maximizes a lower bound on the likelihood of the demonstrations. Experiments comparing I2L to a state-only GAIL baseline show that I2L performs significantly better under dynamics mismatch in several low-dimensional, continuous MuJoCo tasks.

Overall, I enjoyed reading this paper. A few comments:

1. It would be nice to include a behavioral cloning (e.g., BCO) baseline in the experiments. Your point in Section 4 that BC can suffer from compounding errors is well taken, but in my experience, BC can perform surprisingly well on some of the MuJoCo benchmark tasks, even from a single demonstration trajectory. Prior work showing relatively poor results for BC on MuJoCo tasks usually sub-samples demonstrations to intentionally exacerbate the state distribution shift encountered by BC.

2. It would be nice to discuss potential failure cases for I2L. For example, how dependent is the method on the diversity of trajectories \tau generated by the agent in line 5 of Algorithm 1? Are there conditions in which training the critic network to approximate the Wasserstein metric is harder than prior methods like Stadie et al. (2017) and Liu et al. (2018)?

**Experience Assessment:**

I have read many papers in this area.

**Review Assessment: Checking Correctness Of Derivations And Theory:**

I assessed the sensibility of the derivations and theory.

**Review Assessment: Checking Correctness Of Experiments:**

I assessed the sensibility of the experiments.

**Review Assessment: Thoroughness In Paper Reading:**

I read the paper at least twice and used my best judgement in assessing the paper.

---

> ### Author Response · Authors · 2019-11-14
> **Response to AnonReviewer2. Thank you for your comments!**
>
>
> 1- Regarding “BCO baseline”
>
> We have now added Appendix 7.6 comparing I2L to BCO. Since the BCO code is not publicly available, we implemented the algorithm ourselves to the best of our ability using the description provided in the paper. We implement the BCO(alpha) version from the paper since it is shown to be better than vanilla BCO. Appendix 7.6 provides background details on BCO and the results. We observe the barring two situations (Ant with no dynamics mismatch, and Ant with 2x density), BCO is unsuccessful in learning high-return policies. This is potentially due to the difficulties in learning a robust inverse dynamics model, and the compounding errors problem inherent to BC. Similar performance for BCO is also reported by [1], in their Figure 3.
>
> The fact that BCO works even on Ant is surprising enough. We thank the reviewer for pointing this out!
>
> [1] Generative Adversarial Imitation from Observation, Torabi et. al.
>
>
> 2- Regarding “limitations of I2L”
>
> Ensuring sufficient diversity in the trajectories generated by the imitator agent (Line 5, Algorithm 1) might be important in certain situations. This is because these trajectories form the candidate set from which entries are added to the priority-queue buffer, based on the "score" as defined in Section 3. Consider a degenerate case where the agent gets stuck in a local optimum, thereby producing similar trajectories from a small part of the state-action space. These trajectories might not have an adequate score (priority) to enter the buffer, and the algorithm may get stuck. The issue can be potentially alleviated by incorporating techniques for efficient exploration for RL agents, such as those used in single-agent (e.g. intrinsic motivation) and population-based exploration methods. For the MuJoCo environments evaluated in our paper, we find that the exploration induced by a standard Gaussian parameterization for the stochastic policy suffices. But we would add this line of thought to our revised paper, discussing it as possible future work.
>
> For training the Wasserstein critic, we do not expect I2L to add any new challenges, beyond those already encountered by prior approaches which use adversarial training.

---

### Author Response · Authors · 2019-11-14
**General response to the reviewers**

We would like to thank the anonymous reviewers for their comments and constructive feedback. We address each reviewer's comments individually and summarize the major additions to the revision here:

1. Added Appendix 7.3 with further details on the buffer B, and its update mechanism
2. Added Appendix 7.4 with some empirical evaluation of the lower bound used in the paper
3. Added Appendix 7.5 with plots to show a reduction in Wasserstein distance between expert-states and buffer-states
4. Added Appendix 7.6 on comparison to BCO baseline
5. Added Appendix 7.7 with missing learning curves

All changes are in blue text in the revision.

---

### Decision · Program_Chairs · 2019-12-19

**Decision:**

Accept (Poster)

**Comment:**

This paper addresses the setting of imitation learning from state observations only, where the system dynamics under which the demonstrations are performed differs from the target environment. The paper proposes to circumvent this dynamics shift with an algorithm whereby the target policy is trained to imitate its own past trajectories, re-ranked based on the similarity in state occupancies as judged by a WGAN critic.

The reviewers found the paper to be clearly written and enjoyable. The paper improved considerably through reviewers feedback. Notably, a behavior cloning from observations (BCO) baseline was added, which was stronger than the authors expected but still helped highlight the strength of the proposed method by comparison. R1 had a particularly productive multiple round exchange, clarifying the description of previous work, clarifying the details of the proposed procedure and strengthening the presentation of empirical evidence.

This work compellingly addresses an important problem, and in its final form is a polished piece of work. I recommend acceptance.